# Direct and indirect costs attributed to alcohol consumption in Brazil, 2010 to 2018

**Mariana Gonçalves de Freitas**[1]*, **Everton Nunes da Silva**[1,2]

**1** Graduate Program in Public Health, University of Brasilia, Brasilia, Federal District, Brazil, **2** Faculty of Ceilandia, University of Brasilia, Brasilia, Federal District, Brazil

* mfreitas.saude@gmail.com

## Abstract

### Introduction

Alcohol consumption is the main risk factor for death and disability in the world population between 15 to 49 years old, is related to more than 200 causes of death, and has an important economic impact on the health and social security systems. In 2016, three million deaths were attributable to alcohol worldwide and 131.4 million DALYs. In Brazil, alcohol consumption per inhabitant was 7.8 liters in 2016; and the prevalence of abusive consumption was 17.1% in 2019.

### Objective

Estimate the costs attributable of alcohol consumption in the Brazilian population 18 years-old and over, in the period 2010 to 2018.

### Methods

This is a prevalence-based cost-of-illness study, with a top-down and retrospective approach, including direct costs (hospital and outpatient) and indirect costs (absenteeism from work) related to alcohol consumption. A list of diseases and conditions for which alcohol is a risk factor was used, and the Population Attributable Risk (PAR) was calculated to estimate what portion of the cost of these diseases is attributable to alcohol consumption. Prevalence was calculated by sex and level of alcohol consumption, based on data from the 2019 National Health Survey. Relative risk data were identified by disease/injury and level of daily consumption obtained from the Global Burden Disease study in 2017. The cost data used are from the Brazilian Unified Health System and social security system. All costs were adjusted for inflation for the period and converted to purchasing power parity.

### Results

Prevalence data revealed that 73.6% of the Brazilian population reported not consuming alcoholic beverages, which included 62.9% of men and 83.0% of women. The lowest values for the PAR were found at the consumption range of 60 grams per day. Although the 12 grams per day consumption category is the least in terms of the quantity of alcohol consumed, it is one of the main ones in terms of PAR, given the higher prevalence of consumption. The total

**Data Availability Statement:** All relevant data are within the manuscript and its Supporting information files. All data used in this study are from official sources and are available. Data on admission and outpatient care are from the

Ministry of Health and are available at http://www2.datasus.gov.br/DATASUS/index.php?area=02. Absenteeism data are part of the National Institute of Social Security database and were obtained by Brazilian legislation on access to information.

**Funding:** The authors received no specific funding for this work.

**Competing interests:** NO author have competing interests.

cost attributable to alcohol between 2010 and 2018 was Int$ 1,487,417,115.43, of which Int$ 737,834,696.89 was for hospital expenses, Int$ 416,052,029.75 for outpatient care, and Int$ 333,530,388.79 due to absenteeism from work.

## Conclusion

Few comprehensive studies of alcohol-related costs have been developed, which suggests a knowledge gap in Brazil and worldwide and indicates the need for more research in this area. Understanding the economic impact of alcohol consumption is essential to help measure this public health problem in all its aspects and encourage implementation of public policies.

## Introduction

Alcohol consumption is an important public health problem today. In the world population 15 to 49 years old, alcohol is the main risk factor for death and disability [1]. Alcohol consumption is related to more than 200 causes of death, including injuries, cancers, circulatory diseases, and mental health disorders [1, 2]. A study estimated that 3 million (2.6–3.6 95% CI) of deaths were attributable to alcohol in 2016 worldwide and 131.4 million (119.4–154.4 95% CI) disability adjusted life years (DALYs) [3]. This accounted for 5.3% of all deaths and 5% of all DALYs that year [3]. Considering the age group of 20 to 39 years, 13.5% of deaths are attributable to alcohol [2]. A prospective study conducted in Cuba revealed that weekly alcohol consumption was positively associated with premature mortality from all causes, and the risk of death from any cause increased by 10% for each bottle of rum (the main drink consumed in the country) added weekly. The main causes of excessive mortality were cancers, vascular diseases and external causes [4].

There are several mechanisms by which alcohol consumption causes harms to drinkers. First, it is associated with deterioration of organs and tissues, causing diseases such as cancer, liver and heart diseases. Second, it causes alcohol dependence, which is associated with loss of self-control over volume and frequency of alcohol use. This condition also involves mental disorders, such as depression and psychoses. Third, it leads to intoxication, increasing the propensity of getting into dangerous situations under psychoactive effect [2].

In the world population, the total consumption of alcohol among people over 15 years of age increased from 5.5 liters of pure alcohol per inhabitant in 2005 to 6.4 liters in 2010 and remained at this level in 2016. The highest levels of consumption occur in Europe, corresponding to 9.8 liters of pure alcohol per inhabitant [2]. Projections by the World Health Organization (WHO) predict an increase in alcohol consumption to 7.0 liters in 2025 due to the upward trend in the Americas, Asia, and the Western Pacific [2]. The prevalence of heavy episodic drinking (five or more doses for men and four or more for women in a single episode, in the last 30 days) among drinkers aged 15 years and over is 39.5% worldwide, with 40.5% in the Americas, 50.2% in Africa, 42.6% in Europe, and 10.4% in the United Arab Emirates [2].

The economic wealth of countries is associated with alcohol consumption. Globally, richer countries have a higher prevalence of drinkers and higher alcohol consumption per capita, except for the African, which presents the opposite profile. Wealthier countries also have the lowest unrecorded per capita alcohol consumption [2]. Economic development is also an important factor in the consequences of alcohol consumption. Less economically developed countries will have more alcohol-related harm, even if they have the same level of consumption

as other more developed countries, possibly due to difficulty in accessing health services or social exclusion [1, 5].

In Brazil, alcohol consumption per inhabitant was 7.8 liters in 2016, ranging from 13.4 for men and 2.4 for women [2]. In 2013, the prevalence of alcohol abuse in Brazil was 13.7%, ranging from 21.6% of men to 6.6% of women [6]. The results of the National Health Survey in 2019 found an increase in the prevalence of abusive consumption, rising to 17.1% in the total population (26.0% of men and 9.2% of women) [7]. A study conducted by Sandoval et al. with data from a telephone survey conducted in Brazilian capitals identified a prevalence of alcohol consumption at 41%, with 70% of drinkers having regular weekly consuming, 46% presenting heavy episodic consumption, and 28% reporting driving a vehicle after consuming alcoholic beverages [8]. The analysis demonstrated that consumption is highest among men, youth, people with higher education, and individuals who are single or divorced. Variability in the profile of alcohol consumption was observed between the capitals [8].

With an increase in the prevalence of alcohol consumption in Brazil and the world, more knowledge about its economic burden is urgently needed, especially in low- and middle-income countries. These countries tend to suffer most from the scarcity of resources available for health and social security systems. About 80% of the world's health expenditure occurs in high-income countries, which correspond to 20% of the world's population [9]. The disease burden (mortality, disability, and comorbidities) has been well documented in the literature; however, information about the economic burden of alcohol consumption is still scarce. The estimates available are from high-income countries, such as Canada, Portugal, United Kingdom, and USA [10–13]. The estimates for Brazil refer to data from the late 2000s and did not include estimates of indirect costs [14].

In 2010, the WHO published the Global Strategy to reduce the harmful use of alcohol, which still guides the political actions of countries and regions [15]. This document defines best practices, such as reducing the availability of alcohol, regulating alcoholic beverage advertising, and implementing pricing policies [15]. Actions aimed at reducing consumption and harmful effects of alcohol are in line with the Sustainable Development Goals and the Agenda 2030 goals [2, 16].

The objective of this study is to estimate the direct costs (related to health care) and indirect costs (related to absenteeism in the labor market) attributed to alcohol consumption in Brazil between 2010 and 2018, including the diseases or health conditions for which alcohol is a risk factor.

## Method

This is a prevalence-based cost-of-illness study, with a top-down and retrospective approach. The societal perspective was adopted, considering direct costs (outpatient and hospital) and indirect costs (absenteeism) related to alcohol consumption in the Brazilian population age 18 years old or over [17]. Costs were estimated per year for the period 2010 to 2018. Cost estimates were based on national [17] and international [18, 19] methodological recommendations.

To calculate the costs, a list of diseases and conditions for which alcohol is a risk factor was employed [20]. From this list, 21 causes of diseases and injuries were selected. The alcohol use disorders (includes ICD-10 codes: F10, G31.2, X45–X45.9, X65–X65.9, Y15–Y15.9) are fully attributable to its consumption, which accounts for all costs associated with this group. For the other groups of causes, the Population Attributable Risk (PAR) was calculated to estimate what portion of the cost of these diseases is attributable to alcohol consumption.

The PAR was calculated using the following formula:

$$PAR = P(RR - 1)/P(RR - 1) + 1$$

In which, P is the prevalence of alcohol consumption in the Brazilian population age 18 years and over and RR is the relative risk of developing a certain disease or occurrence of an injury in individuals who consume alcohol, according to the level of alcohol consumption, versus nondrinkers.

## Prevalence

The prevalence of alcohol consumption in the adult Brazilian population was estimated based on data from the National Health Survey (PNS), conducted in 2019 [7]. The PNS is a population-based and nationally representative survey conducted by the Ministry of Health and by the Brazilian Institute of Geography and Statistics (IBGE). Three questionnaires were applied during this research. The first was the household questionnaire that captured information about the residence, and the second questionnaire was applied to each resident of the household with questions about income, education, etc. For the third questionnaire that collected information about individuals' lifestyle and morbidity, a single resident aged 15 and over was selected randomly with equal probability. The sampled 108,525 households included representation from all Federative Units, capitals, and metropolitan regions [7].

For this study, two questions from module P–Lifestyles of the individual resident questionnaire were asked: Question P028 –How many days a week do you usually consume any alcoholic beverage? and P029 –In general, on the day you drink, how many servings of an alcoholic beverage do you consume? (1 serving of alcoholic beverage is equivalent to 1 can of beer, 1 glass of wine, or 1 shot of cachaça, whiskey, or any other distilled alcoholic beverage) [21]. From these questions, the *daily consumption* variable was established from the multiplication of the questions divided by 7 to obtain the daily serving. This value was then multiplied by 12 to obtain the daily dose in grams. For this consumption pattern, the standard serving of pure alcohol was used, such as 10 to 12 grams of alcohol per serving.

Seven alcohol consumption levels were created, with intervals every 12 grams, until reaching the consumption of 72 grams or more per day. The prevalence was calculated separately for each sex and for both sexes, for individuals age 18 years and over. Prevalence data were calculated using Stata® version 14.0 software.

## Relative risk

Relative risk (RR) data used in this study were obtained from the publication *Global Burden of Disease* (GBD) 2017 [20]. RR data are presented by level of consumption (every 12 grams) and ICD-10 code. Chart 1 lists the diseases and conditions in which alcohol is identified as a risk factor for morbidity and mortality.

The initial list of alcohol-related disease and injury codes obtained in the GBD publication was used to define the main codes for each disease and injury and their sequelae as well as the data used. RR data were stratified by consumption ranges, for each 12 grams of daily consumption. The records of intracerebral hemorrhage were analyzed according to sex. The RR confidence interval was also used.

## Costs

For the population 18 and over, direct costs of outpatient care (amount approved for outpatient services) obtained from the SIA (Outpatient Information System) and costs of hospital

admissions (total amount) obtained from the Hospital Information System (SIH-SUS) were considered. The SIA and SIH-SUS data are public data and were obtained from DATASUS and processed on Tabwin software.

Indirect costs related to removal from the formal labor market due to alcohol consumption (or attributed to alcohol consumption) were also included. These costs were calculated based on data about the sick pay benefit from the National Social Security Institute (INSS), obtained through the Access to Information Law–LAI. The list of disease and injury codes used for indirect cost calculations was the same as described in Table 1.

Data on costs of sick pay were provided by ICD-10 code, per year, by number of benefits granted, duration of benefit, and initial monthly income (IMI) per benefit. The total cost per ICD-10 per year was calculated using the following formula:

$$\text{Average daily income (ADI)} = \text{Initial monthly income (IMI)}/30$$

$$\text{Total cost} = \Sigma[(\text{ADI} * \text{duration of benefit (in days)}) * \text{total number of benefits}]$$

Finally, the alcohol-attributable costs for each information system were calculated using the formula:

$$\text{Attributable costs} = \text{PAR} * \text{Total cost ICD (for each information system)}$$

All cost values were corrected for inflation for the period by the National Consumer Price Index—CPI and by purchasing power parity (conversion factor equal to 2.2), with reference to the year 2018 [22]. The values are presented in international dollars (Int$).

**Table 1. Groups of diseases and conditions included in the study.**

| Illness/injury | ICD-10 Codes |
| --- | --- |
| 1- Tuberculosis and tuberculosis sequelae | A15–A19.9, B90–B90.9 |
| 2- Lower respiratory infections | J09– J22.9, J85.1 |
| 3- Cancer of the esophagus | C15–C15.9 |
| 4- Liver cancer due to alcohol use | C22–C22.9 |
| 5- Laryngeal cancer | C32–C32.9 |
| 6- Breast Cancer | C50–C50.929 |
| 7- Colon and rectum cancer | C18–C21 |
| 8- Lip and oral cavity cancer | C00–C08.9 |
| 9- Nasopharyngeal cancer | C11–C11.9 |
| 10- Other pharyngeal cancers | C09–C10.9, C12–C13.9 |
| 11- Ischemic heart disease | I20–I25.9 |
| 12- Intracerebral hemorrhage | I60–I62.9, I69.0–I69.2 |
| 13- Atrial fibrillation and flutter | I48–I48.9 |
| 14- Cirrhosis and other chronic liver diseases due to alcohol use | K70–K70.9 |
| 15- Pancreatitis | K85–K86.9 |
| 16- Epilepsy | G40–G41.9 |
| 17- Traffic accidents | V01–V99 |
| 18- Accidental injuries | W00–X29.9, X40–X40., X58.99, X43–X43.9, X46–X48.9 |
| 19- Intentionally self-inflicted injury and sequelae | X60–X64.9, X66–X84.9, Y87.0 |
| 20- Interpersonal violence and sequelae | X85–Y08.9, Y87.1, Y87.2 |
| 21- Alcohol use disorders | F10, G31.2, X45–X45.9, X65–X65.9, Y15–Y15.9 |

## Results

Prevalence data reveal that 73.61% of the Brazilian population reported not consuming alcoholic beverages, including 62.91% of men and 83.04% of women (Table 2). The distribution of the prevalence of daily alcohol consumption in grams showed that 14.70% of the Brazilian population consumed an average of 12 grams of alcohol per day, which was 18.04% of men and 11.76% of women. The prevalence of drinking according to consumption category decreased as the daily intake of alcohol in grams increased, and 1.18% of the Brazilian population consumed 72 grams of alcohol per day (Table 2).

Table 3 reveals that the lowest values for the PAR were found in the consumption range of around 60 grams per day. The lowest prevalence of alcohol consumption occurs in this group, with values of 0.57%. Although the consumption category of 12 grams per day is the smallest in terms of the amount of alcohol consumed, it is one of the main ones in terms of PAR, due to the higher prevalence of consumption (14.7%). The group that consumes 72 grams (and more) per day also displays a high PAR, due to the risk of such an extensive drinking pattern, even if the prevalence in this group is low (1.18%).

Table 4 presents the costs attributable to alcohol during the study period (2010 to 2018) and the average costs for the period. The total cost attributable to alcohol between 2010 and 2018 was Int$ 1,487,417,115.43, which was Int$ 737,834,696.89 for hospitalizations, Int$ 416,052,029.75 for outpatient procedures, and Int$ 333,530,388.79 for absenteeism.

The injuries that contributed most to hospital costs were alcohol use disorders (38.6% of costs), unintentional injuries (12.4%), and traffic accidents (8.4%). The main injuries and illnesses that generated outpatient costs were breast cancer (46.1%), alcohol use disorders (15.7%), and colon and rectal cancer (13.4%). The average costs during the period due to absenteeism were mostly from alcohol use disorders (57.9%) as well as breast cancer (12.9%) and tuberculosis (8.3%) (Table 4).

Fig 1 illustrates the costs related to alcohol according to the type of cost between 2010 and 2018. The hospitalization was the most expensive type of cost throughout the period. From 2010 to 2012, outpatient costs were in the order of Int$ 60 million per year, starting in 2013, outpatient and absenteeism costs remain very close until 2018, around Int$ 40 million. The three types of costs added together was, on average, Int$ 165,268,568.38 per year due to the use of alcohol.

Fig 2 displays the proportion of expenses attributable to alcohol in relation to the total cost for each disease. The diseases with the highest proportion of attributable cost are cirrhosis, other pharyngeal cancers, nasopharyngeal cancer, followed by alcohol use disorders which are 100% attributable to drinking. The diseases with the lowest proportion are lower respiratory infections, intracerebral hemorrhage for women, unintentional injuries.

**Table 2. Prevalence of alcohol intake by consumption group and sex, Brazil, 2019.**

| Daily consumption of alcohol in grams | Male | Females | Everyone |
|---|---|---|---|
| 0 | 0.6291 | 0.8304 | 0.7361 |
| 12 | 0.1804 | 0.1176 | 0.1470 |
| 24 | 0.0940 | 0.0312 | 0.0606 |
| 36 | 0.0484 | 0.0107 | 0.0283 |
| 48 | 0.0174 | 0.0043 | 0.0105 |
| 60 | 0.0102 | 0.0017 | 0.0057 |
| $\geq$ 72 | 0.0205 | 0.0041 | 0.0118 |

Source: Compiled by the authors based on data from PNS 2019.

**Table 3. Population Attributable Risk by ICD-10 and level of consumption.**

| ICD-10 | Level of Consumption | | | | | |
|---|---|---|---|---|---|---|
| | 12 g/day | 24 g/day | 36 g/day | 48 g/day | 60 g/day | 72 g/day |
| Tuberculosis | 0.01463 | 0.03118 | 0.02907 | 0.01586 | 0.01124 | 0.02873 |
| Lower respiratory infections | 0.00191 | 0.00157 | 0.00181 | 0.00133 | 0.00129 | 0.00419 |
| Esophageal cancer | 0.03022 | 0.02746 | 0.02254 | 0.01246 | 0.00821 | 0.01931 |
| Liver cancer due to alcohol use | 0.00975 | 0.00841 | 0.00633 | 0.00324 | 0.00212 | 0.00498 |
| Laryngeal cancer | 0.01733 | 0.01809 | 0.01480 | 0.00846 | 0.00648 | 0.01695 |
| Breast cancer | 0.02438 | 0.01955 | 0.01211 | 0.00463 | 0.00257 | 0.00559 |
| Colon and rectum cancer | 0.01134 | 0.00937 | 0.00666 | 0.00338 | 0.00266 | 0.00722 |
| Lip and oral cavity cancer | 0.04129 | 0.04281 | 0.03577 | 0.02048 | 0.01552 | 0.04354 |
| Nasopharyngeal cancer | 0.05172 | 0.04838 | 0.03772 | 0.02119 | 0.01573 | 0.04015 |
| Other pharyngeal cancers | 0.06488 | 0.05406 | 0.04122 | 0.02257 | 0.01666 | 0.04253 |
| Hypertensive heart disease | 0.00672 | 0.01873 | 0.01337 | 0.00641 | 0.00400 | 0.01005 |
| Atrial fibrillation and flutter | 0.00961 | 0.00788 | 0.00602 | 0.00327 | 0.00234 | 0.00627 |
| Cirrhosis and other chronic liver diseases due to alcohol use | 0.03449 | 0.06009 | 0.06046 | 0.03713 | 0.02918 | 0.09044 |
| Pancreatitis | 0.01062 | 0.01363 | 0.01315 | 0.00747 | 0.00689 | 0.02640 |
| Epilepsy | 0.02536 | 0.02094 | 0.01629 | 0.00907 | 0.00671 | 0.01716 |
| Traffic accident | 0.02340 | 0.01316 | 0.00808 | 0.00383 | 0.00259 | 0.00647 |
| Accidental injuries | 0.01306 | 0.00925 | 0.00473 | 0.00191 | 0.00126 | 0.00313 |
| Self-inflicted injury | 0.01549 | 0.01375 | 0.01053 | 0.00569 | 0.00417 | 0.01082 |
| Interpersonal violence | 0.01861 | 0.01528 | 0.00967 | 0.00414 | 0.00257 | 0.00605 |
| Intracerebral hemorrhage—male | 0.01212 | 0.01500 | 0.01478 | 0.00791 | 0.00714 | 0.01952 |
| Intracerebral hemorrhage—female | 0.00363 | 0.00342 | 0.00359 | 0.00263 | 0.00164 | 0.00520 |
| key | | | | | | |
| 1 | lowest PAR | | | | | |
| 2 | | | | | | |
| 3 | | | | | | |
| 4 | | | | | | |
| 5 | | | | | | |
| 6 | highest PAR | | | | | |

Fig 3 reveals that for most causes, the main costs are for hospitalization, followed by outpatient care, and then costs for sick pay benefits. For breast cancer, outpatient expenses are the greatest contributor to the total cost attributable to alcohol, as well as for nasopharyngeal cancer, other pharyngeal cancer, and colon and rectum cancer.

Hospital costs from external causes (traffic accidents, interpersonal violence, accidental injuries, and self-harm) are high when comparable to other costs for these causes, as well as lower respiratory infections. The hospital component represents more than 96% of the costs for these causes. Absenteeism-related costs are high for hypertensive heart disease, tuberculosis, epilepsy, and alcohol use disorders.

For more information and data about costs attributable of alcohol between 2010 to 2018, please see the supporting information in S1 Table-Costs-2010 until S9 Table-Costs-2018. For data about Relative Risk and Confiance Interval, please see the file S10 Table-PAR-Prevalence-RR.

## Discussion

The study estimated that the average cost attributable to alcohol per year exceeds Int$165 million. The budget of the Brazilian National Health System in 2018 was Int$ 55.39 billion, of

**Table 4. Total and average costs attributable to alcohol by type of cost and ICD, Brazil, 2010 to 2018.**

| CID-10 | Total Costs attributed to alcohol | Total cost attributable to alcohol by type of cost 2010–2018 | | | Total mean cost attributed to alcohol * | Total mean cost attributable to alcohol by type of cost 2010–2018* | | |
|---|---|---|---|---|---|---|---|---|
| | | Hospital | Outpatient | Absenteeism | | Hospital | Outpatient | Absenteeism |
| Tuberculosis | 47.643.042.92 | 19,427,446.88 | 436,749.88 | 27,778,846.17 | 5,293,671.44 | 2,158,605.21 | 48,527.76 | 3,086,538.46 |
| Lower respiratory infections | 25,882,645.34 | 25,040,424.96 | 99,926.29 | 742,294.10 | 2,875,849.48 | 2,782,269.44 | 11,102.92 | 82,477.12 |
| Esophageal cancer | 38,407,392.14 | 15,905,228.63 | 17,547,177.54 | 4,954,985.97 | 4,267,488.02 | 1,767,247.63 | 1,949,686.39 | 550,554.00 |
| Liver cancer due to alcohol use | 3,465,637.57 | 2,061,433.67 | 594,547.32 | 809,656.57 | 385,070.84 | 229,048.19 | 66,060.81 | 89,961.84 |
| Laryngeal cancer | 19,447,320.82 | 7,668,849.74 | 9,032,795.37 | 2,745,675.71 | 2,160,813.42 | 852,094.42 | 1,003,643.93 | 305,075.08 |
| Breast cancer | 266,990,559.40 | 32,100,565.32 | 191,885,190.64 | 43,004,803.44 | 29,665,617.71 | 3,566,729.48 | 21,320,576.74 | 4,778,311.49 |
| Colon and rectum cancer | 96,011,570.66 | 28,750,737.97 | 55,665,174.73 | 11,595,657.96 | 10,667,952.30 | 3,194,526.44 | 6,185,019.41 | 1,288,406.44 |
| Lip and oral cavity cancer | 76,123,618.64 | 33,840,222.65 | 31,661,681.92 | 10,621,714.06 | 8,458,179.85 | 3,760,024.74 | 3,517,964.66 | 1,180,190.45 |
| Nasopharyngeal cancer | 10,248,380.39 | 1,962,825.24 | 6,182,415.16 | 2,103,140.00 | 1,138,708.93 | 218,091.69 | 686,935.02 | 233,682.22 |
| Other pharyngeal cancers | 47,090,994.23 | 9,950,902.47 | 31,114,047.20 | 6,026,044.56 | 5,232,332.69 | 1,105,655.83 | 3,457,116.36 | 669,560.51 |
| Hypertensive heart disease | 3,879,420.55 | 1,232,206.26 | 363,581.22 | 2,283,633.06 | 431,046.73 | 136,911.81 | 40,397.91 | 253,737.01 |
| Atrial fibrillation and flutter | 2,296,943.77 | 1,673,350.56 | 22,738.36 | 600,854.85 | 255,215.97 | 185,927.84 | 2,526.48 | 66,761.65 |
| Cirrhosis and other chronic liver diseases due to alcohol use | 68,381,611.06 | 56,861,331.80 | 759,494.65 | 10,760,784.61 | 7,597,956.78 | 6,317,925.76 | 84,388.29 | 1,195,642.73 |
| Pancreatitis | 11,912,645.95 | 8,184,207.27 | 1,623,535.18 | 2,104,903.50 | 1,323,627.33 | 909,356.36 | 180,392.80 | 233,878.17 |
| Epilepsy | 20,991,076.14 | 9,604,852.94 | 2,479,606.73 | 8,906,616.48 | 2,332,341.79 | 1,067,205.88 | 275,511.86 | 989,624.05 |
| Traffic accident | 62,903,498.49 | 62,185,430.41 | 210,744.69 | 507,323.39 | 6,989,277.61 | 6,909,492.27 | 23,416.08 | 56,369.27 |
| Accidental injuries | 92,341,736.73 | 91,674,976.10 | 248,752.60 | 418,008.03 | 10,260,192.97 | 10,186,108.46 | 27,639.18 | 46,445.34 |
| Self-inflicted injury | 1,855,511.32 | 1,776,470.08 | 6,832.33 | 72,208.90 | 206,167.92 | 197,385.56 | 759.15 | 8,023.21 |
| Interpersonal violence | 17,109,205.58 | 16,348,062.35 | 82,003.08 | 679,140.15 | 1,901,022.84 | 1,816,451.37 | 9,111.45 | 75,460.02 |
| Intracerebral hemorrhage—male | 23,628,420.10 | 20,155,874.94 | 502,652.81 | 2,969,892.35 | 2,625,380.01 | 2,239,541.66 | 55,850.31 | 329,988.04 |
| Intracerebral hemorrhage—female | 7,151,925.27 | 6,368,289.90 | 144,657.92 | 638,977.46 | 794,658.36 | 707,587.77 | 16,073.10 | 70,997.50 |
| Alcohol use disorders | 543,653,958.36 | 285,061,006.75 | 65,387,724.12 | 193,205,227.49 | 60,405,995.37 | 31,673,445.19 | 7,265,302.68 | 21,467,247.50 |
| Total | 1,487,417,115.43 | 737,834,696.89 | 416,052,029.75 | 333,530,388.79 | 165,268,568.38 | 81,981,632.99 | 46,228,003.31 | 37,058,932.09 |

*mean costs in the period

Values presented in Int$

which Int$ 22.84 billion was spent on hospital and outpatient care [23]. The hospital and out-patient costs attributable to alcohol estimated in the study was Int$ 128.21 million and corresponds to 0.56% of this cost in 2018.

The data on the prevalence of alcohol consumption in Brazil in 2019 indicate that alcohol consumption is lower in women, which is in line with international estimates [2]. The lowest range of alcohol consumption (12 g/day) had to the greatest impact on the PAR. These findings are relevant and corroborated by a study by Griswold et al., which demonstrates that there is no level of alcohol consumption that does not harm health, even in drinkers who consume one drink a day [24]. Increasing the amount of alcohol consumed increases the risk of mortality from all causes, including cancers [24]. A study by Wood et al. analyzed the safe limits for alcohol consumption in 559,912 referred drinkers in 83 prospective studies. The

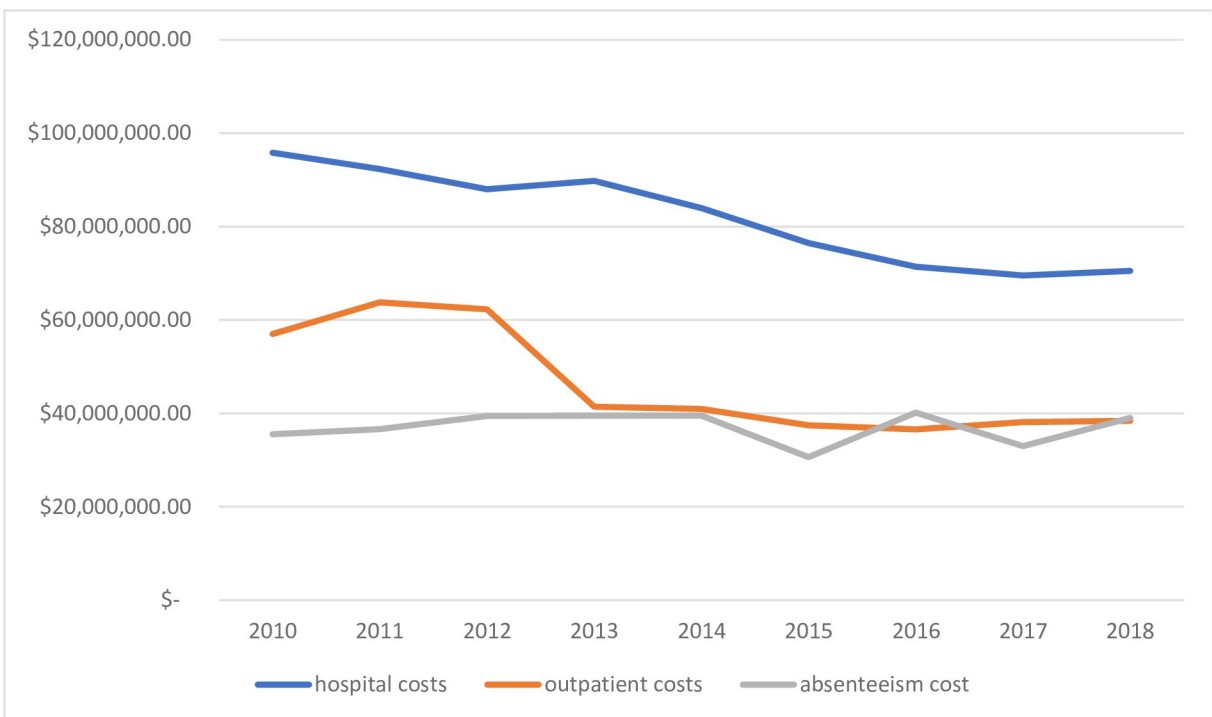

**Fig 1. Evolution of total cost attributable to alcohol per year, by type of cost Brazil, 2010 to 2018.**

findings also reveal that there is no safe limit for consumption that minimizes the risk of illness and death in all subtypes of cardiovascular diseases, except for myocardial infarction [25]. The risks associated with alcohol consumption outweigh any possible protective effects for cardiovascular diseases that have been suggested by previous studies [24]. The higher the level of alcohol consumption, the shorter the life expectancy. At 40 years of age, the life expectancy is reduced up to 5 years for individuals who consume more than 350 grams of alcohol per week [25].

A study by Coutinho et al. estimated the total costs for all diseases related to the risky consumption of alcohol at US$8,262,762.00 (US$4,413,670.00 and US$3,849,092.00 for outpatient and inpatient expenses, respectively), in Brazil in 2014 [14]. The results of our research are 14.5 times higher than the study by Coutinho et al., if we consider only the costs arising from hospital and outpatient care. This is due to the wide range of alcohol-related illnesses and injuries that were included in our study. The outpatient component is the main cost in Coutinho's study, while our study has hospital expenses as its main cost component. This difference is due to the list of diseases and conditions adopted in each study. Coutinho et al. chose 8 ICD-10 codes for their study, 5 of which are related to cancer, which has a strong impact on outpatient care. Furthermore, the study by Coutinho et al. adopted an RR cut-off greater than 1.2 for the diseases or injuries included in the analysis, without stratification by alcohol consumption. Our study included any RR value reported in the GBD study, and stratified daily alcohol consumption every 12 g. These two factors tend to capture disease costs more broadly. Another positive point of our study is that we estimate indirect costs (absenteeism), even if restricted to the formal labor market. The two studies corrected the calculated value of the cost according to the PPP (Purchasing Power Parity).

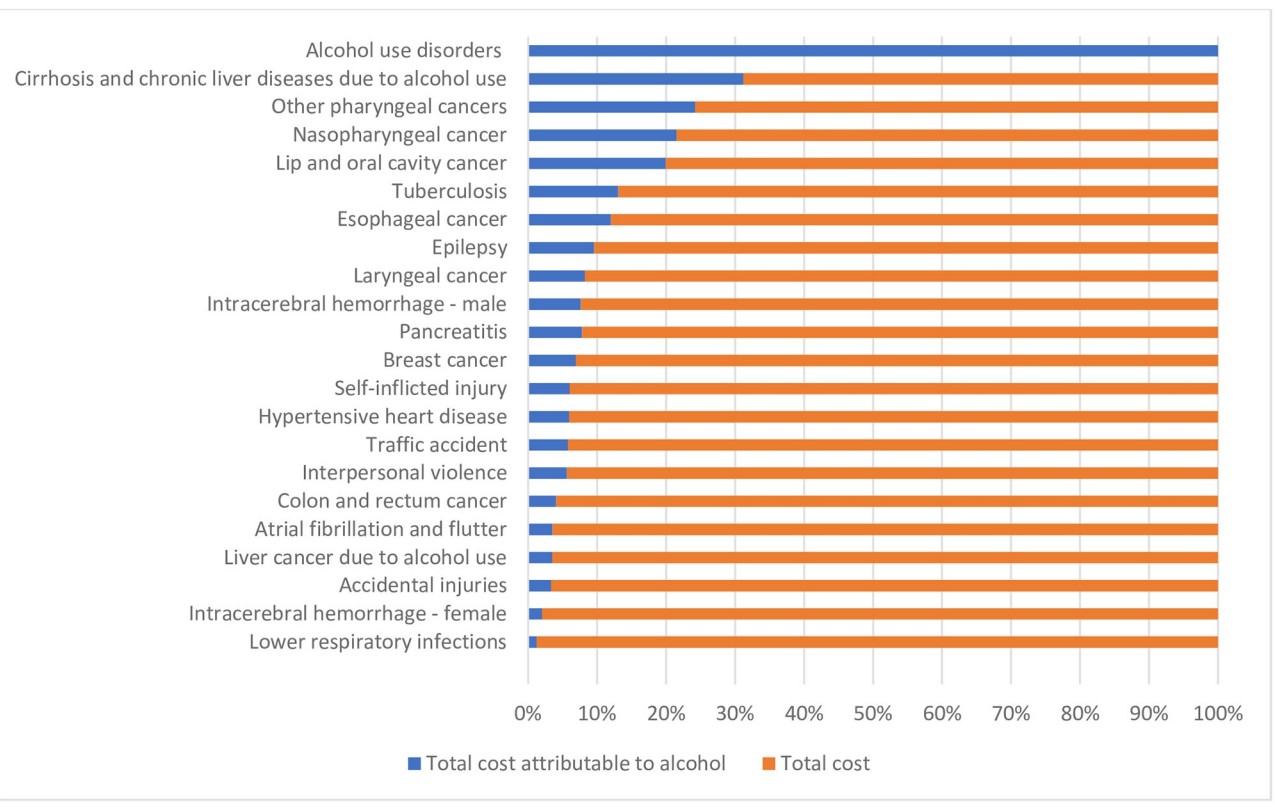

**Fig 2. Proportion of total cost attributable to alcohol according to ICD-10, Brazil, 2010–2018.**

A cost study conducted in Canada estimated the cost of alcohol at CA\$14.6 billion in 2014 [10]. Between 2007 and 2014, the per capita cost of alcohol-related health care increased by 25.9%. The main alcohol-related cost was lost productivity (40.4), followed by health care (28.9%), criminal justice (21.5%), and other direct costs (9.2%). In our study, indirect costs accounted for 22.4% of the estimated total cost, which is about half the proportion estimated in the Canadian study. However, we only considered absenteeism in the formal labor market. In Brazil, an estimated 41.6% of workers work in the informal sector, according to IBGE data for 2019, and the rate of this informality varies according to worker's race, sex, and education level [26]. The limitation of accessing informal workers leads to an underestimation of the indirect costs of alcohol in Brazil. Furthermore, the Canadian study used a broader set of data sources for indirect costs, including absenteeism, presenteeism, and early death.

Data from demographic and health statistics for Portugal in 2005 found a cost of €95.1 million in hospital admissions for alcohol-related illnesses and injuries and €95.9 million in outpatient care, totaling €191 million, which represents 1.3% of the country's total health care costs and 0.13% of Portugal's GDP [11]. Our study estimated that the cost of alcohol consumption corresponded to 0.56% of hospital and outpatient expenses in the public health system and to 0.23% of total public health expenditure. Thus, the proportion the total health expenditure attributed to alcohol was 5.6 times higher in the Portuguese case than in the Brazilian case. In addition to the methodological issues and databases used, this difference may also be related to what the two health systems provide, which affects the denominator of the proportion being compared. In the Brazilian case, the health system adopts the principle of integrality, guaranteeing free access to all health care throughout life [27]. Thus, the set of individual and

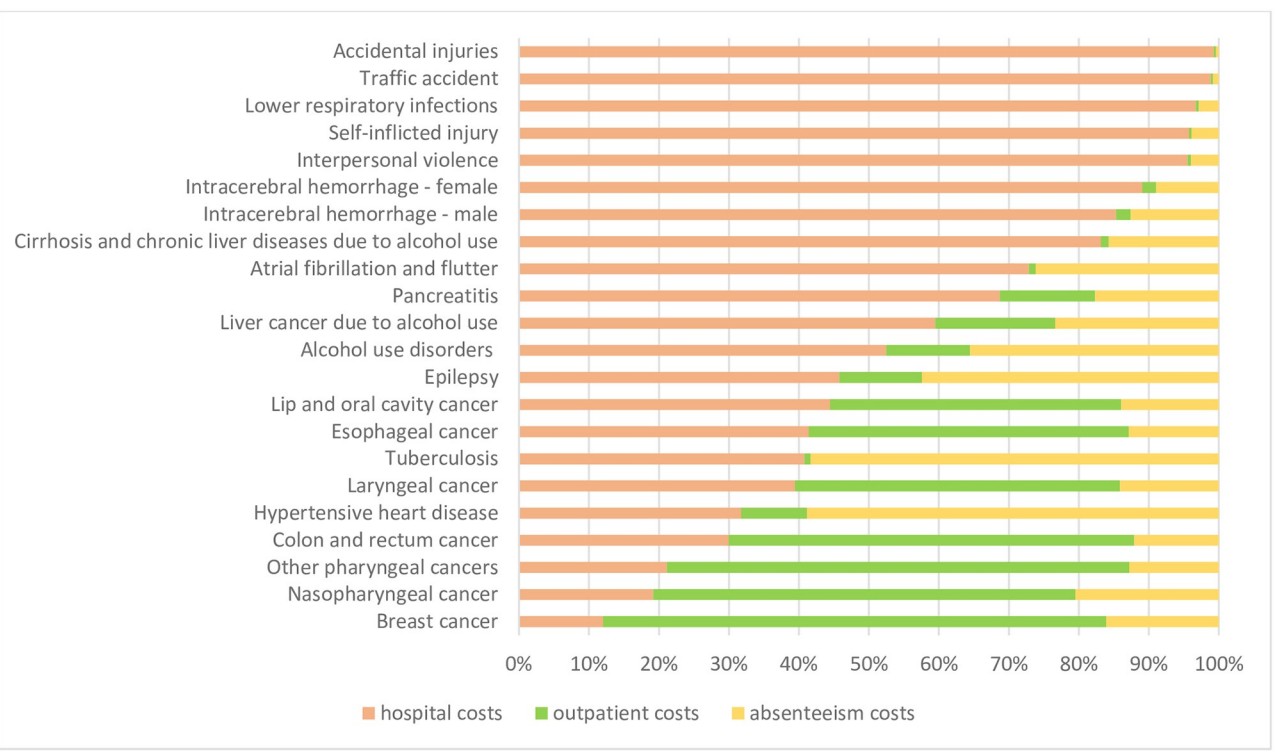

**Fig 3. Proportion distribution of cost attributable to alcohol according to ICD and the Brazil information system, 2010–2018.**

collective actions tends to be greater in Brazil than in Portugal, which may partly explain the lower proportion of alcohol related costs of the total expenditure in Brazil.

The estimated total cost of alcohol in Sri Lanka was US$ 885.86 million, with US$ 388.35 million in direct costs (health care costs) and US$ 497.50 million in indirect costs (social security costs). The total cost corresponds to 1% of the country's GDP in 2015 [28]. This study contrasts with the results found in our study, the social security cost component corresponds to 22.4% of the total cost, while in the Sri Lanka survey this component represents 56.2% of the estimated cost [28]. As highlighted above, our absenteeism costs refer only to the formal labor market in Brazil, which may partially explain this different proportion.

A study of the economic burden of alcohol consumption in the United Kingdom estimated the cost of 3 billion pounds for the National Health Service, in 2005 and 2006, which is equivalent to 3.2% of the total expenditure on health. Of this, 939 million were on alcohol use disorders, 374 million spent on patients with liver cirrhosis, and 334 million on motor vehicle accidents [12]. Compared to the results of our study, the main expense for health care (inpatient and outpatient) was also for alcohol use disorders (Int$ 38.9 million), while the second expense was for breast cancer (Int$ 24.9 million), followed by accidental injuries (Int$10.2 million).

Our study has several advantages over the previous Brazilian study and adds information in this field of knowledge. In this study, both healthcare (inpatients and outpatients) and medical leave databases were used for more complete composition of data. Another important advantage is the historical series of the study, which includes nine years of records from all analyzed databases. This series allowed a more consistent average for the period and analysis of possible trends. The research used relative risk data from the publication of Global Burden of Disease (GBD) 2017, which were obtained from systematic reviews. Another important advantage

refers to the extensive list of codes of diseases and conditions used, which includes 21 groups of causes, increasing the range of results compared to a previous study conducted in Brazil.

Nevertheless, this study has some limitations. The healthcare data (SIH-SUS and SIA-SUS) are only for care provided in the public health system. Thus, all care for these causes in private hospitals and outpatient clinics not affiliated with the SUS are not included in this study, which underestimates the results presented. About 25% of the Brazilian population have access to private health insurances, which was not cover by our study. In addition, the amounts presented are those reimbursed by the Ministry of Health, which may not reflect the actual amount incurred by healthcare providers. Another limitation is that the INSS data only records sick leave for formal workers in Brazil. In addition, only leave longer than 15 days is registered with the INSS, in case of shorter leave, the employer is responsible for the costs, and there are no records of these expenses. Moreover, we have not included early retirements and premature deaths attributable to alcohol consumption, which is also an important source of indirect costs. All these limitations tend to reduce the picture of the real economic burden of alcohol in Brazil. Our rationale for not including simulations or adjustments to correct these information system failures is that our objective was to provide real-world estimates based on the best available healthcare information systems. The observed reduction in outpatient and hospital costs after 2012 can be partly attributed to the reduction of psychiatric beds and the change in classification of the mental healthcare registry, including alcohol use disorders, which may indicate that the data are underestimated. Finally, our main source for relative risk from each disease by which alcohol consumption if risk factor did not provide data stratified by sex and age groups. On this basis, we were not able to provide cost estimated by sex and age groups.

Given the full burden of morbidity and mortality and economic impacts associated with alcohol consumption, the Global Strategy to reduce the harmful alcohol use is a tool that helps policy development. Proposed strategies included regulating the sale of alcohol, especially increasing the price and decreasing availability, which are cost-effective ways to reduce alcohol-related harm and are among the best practices [15, 29]. Taxation is an effective mechanism for reducing the prevalence, frequency, and intensity of consumption of this produce. This measure has a greater impact on adolescents and heavy drinkers, reducing consumption and damage caused, and taxation has also been identified as the most effective means of reducing drunk driving as well as crime and absenteeism related to alcohol. The adoption of these measures aimed at reducing consumption and preventing intoxication are effective strategies to reduce the harm caused by alcohol [30]. A strategy carried out in Diadema, São Paulo, to reduce homicides was controlling the availability of alcohol and closing bars after 11 pm, which resulted in a reduction of homicides after the implementation of the policy, confirming that limiting access can reduce problems related to alcohol consumption [31].

Literature review of English-language articles published between 1980 and 2006 identified the broad influence of the alcohol industry on policy formulation [32]. Their influence includes framing political debates, excluding issues contrary to their commercial interests, and managing the commercial interests of industries in the political decision-making space [32].

The WHO indicated that 25.5% of all alcohol consumed worldwide is unregistered and outside government control [2]. Measures such as increased control of the unregistered alcohol sales, regulation of beer advertising (currently not considered an alcoholic beverage for advertising purposes in Brazil), and increased inspection for underage drinking deserve attention and investment. Finally, the complexity of the problem of alcohol consumption in Brazil and in the world is underscored, which has health and economic impacts and requires political interventions and dialogue between the public sphere, the alcohol industry, and civil society.

There are few cost studies related to alcohol produced in Brazil. Thus, the production of knowledge in this area is necessary. Understanding the economic impacts due to alcohol consumption is essential to scale this public health problem in all its facets, in addition to the morbidity and mortality it causes. There is also need for evidence on disease burden of alcohol consumption on other diseases than ones included in our study, including mortality outcome. It is also important to investigate the barriers and facilitators of implementing policies against alcohol consumption among age groups, since they may respond differently depend on the strategy adopted.

## Supporting information

**S1 Table. Table costs 2010.** Costs attributable to alcohol by type of cost and ICD, Brazil, 2010. (PDF)

**S2 Table. Table costs 2011.** Costs attributable to alcohol by type of cost and ICD, Brazil, 2011. (PDF)

**S3 Table. Table costs 2012.** Costs attributable to alcohol by type of cost and ICD, Brazil, 2012. (PDF)

**S4 Table. Table costs 2013.** Costs attributable to alcohol by type of cost and ICD, Brazil, 2013. (PDF)

**S5 Table. Table costs 2014.** Costs attributable to alcohol by type of cost and ICD, Brazil, 2014. (PDF)

**S6 Table. Table costs 2015.** Costs attributable to alcohol by type of cost and ICD, Brazil, 2015. (PDF)

**S7 Table. Table costs 2016.** Costs attributable to alcohol by type of cost and ICD, Brazil, 2016. (PDF)

**S8 Table. Table costs 2017.** Costs attributable to alcohol by type of cost and ICD, Brazil, 2017. (PDF)

**S9 Table. Table costs 2018.** Costs attributable to alcohol by type of cost and ICD, Brazil, 2018. (PDF)

**S10 Table. Relative risk, prevalence of alcohol intake, Population Attributable Risk by disease or injuries and level of consumption.**
(PDF)

## Author Contributions

**Conceptualization:** Mariana Gonçalves de Freitas, Everton Nunes da Silva.

**Data curation:** Mariana Gonçalves de Freitas.

**Formal analysis:** Mariana Gonçalves de Freitas.

**Methodology:** Mariana Gonçalves de Freitas, Everton Nunes da Silva.

**Supervision:** Everton Nunes da Silva.

**Visualization:** Mariana Gonçalves de Freitas, Everton Nunes da Silva.

**Writing – original draft:** Mariana Gonçalves de Freitas, Everton Nunes da Silva.

**Writing – review & editing:** Mariana Gonçalves de Freitas, Everton Nunes da Silva.

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
