## [Decision Letter · Decision Letter 0]

14 Mar 2022

PONE-D-21-23873Direct and indirect costs attributed to alcohol consumption in Brazil, 2010 to 2018PLOS ONE

Dear Dr. de Freitas,

Thank you for submitting your manuscript to PLOS ONE. After careful consideration, we feel that it has merit but does not fully meet PLOS ONE’s publication criteria as it currently stands. Therefore, we invite you to submit a revised version of the manuscript that addresses the points raised during the review process.

We look forward to receiving your revised manuscript.

Kind regards,

Roy Cerqueti, Ph.D.

Academic Editor

PLOS ONE

Journal Requirements:

2. Thank you for stating the following financial disclosure: "Unfunded studies"

3. Please provide additional details regarding participant consent. In the Methods section, please ensure that you have specified (1) whether consent was informed and (2) what type you obtained (for instance, written or verbal). If your study included minors, state whether you obtained consent from parents or guardians. If the need for consent was waived by the ethics committee, please include this information.

Reviewers' comments:

Reviewer's Responses to Questions

**Comments to the Author**

1. Is the manuscript technically sound, and do the data support the conclusions?

Reviewer #1: Yes

Reviewer #2: Yes

2. Has the statistical analysis been performed appropriately and rigorously? 

Reviewer #1: Yes

Reviewer #2: Yes

3. Have the authors made all data underlying the findings in their manuscript fully available?

Reviewer #1: Yes

Reviewer #2: Yes

4. Is the manuscript presented in an intelligible fashion and written in standard English?

Reviewer #1: Yes

Reviewer #2: Yes

5. Review Comments to the Author

Reviewer #1: The manuscript aims to estimate the direct costs and indirect costs attributed to alcohol consumption in Brazil between 2010 and 2018, considering the diseases for which alcohol is a risk factor.

Introduction

- The introduction is complete in describing the health burden of alcohol consumption in the world, the trends in consumption in Brazil and other aspects of the issue. Nevertheless, it would benefit from addressing how alcohol, as a risk factor, is associated to the main health outcomes of excessive consumption.

- Additionally, what are the trends of alcohol consumption over time according to the national telephone survey, which is implemented yearly?

Methods

- The study uses a traditional cost-of-illness approach, using the population attributable risk associated with the prevalence of alcohol consumption and relative risks for different disease outcomes.

- It is not clear if the PAR are calculated for the entire population or if it is separated by age and sex groups. The disaggregation would be important to incorporate the differences in alcohol consumption among different population groups, such as those reported in the introduction (“consumption is highest among men, youth, people with higher education, and individuals who are single or divorced”).

- The PAR are based on a deterministic approach for the outputs. Nevertheless, estimations of the health and economic burden of disease or risk factors could use a probabilistic approach to incorporate the uncertainty of the inputs in the final results. Has this option been considered?

- Regarding the prevalence of alcohol consumption, why was the data from the 2013 National Health Survey included in the analysis? It may provide possible time changes in consumption in the period analyzed.

- Why were the early retirements caused by diseases not included in the indirect costs? This data is usually available from the same sources as the sick leaves.

Results

- It is important to include the confidence intervals for the estimates, both in the tables and in the text.

- Would it be possible to express the standard deviation or standard error (or the IC 95%) for the prevalences, together with the IC for the relative risks?

Discussion

- Lines 290-304: are the results of this study on the hospitalization costs for cancer comparable to the study by Coutinho et al?

- Line 310: The estimation of absenteeism is also underestimated because the sick pay leaves only include periods of 15 days or more away from the job paid by the INSS in Brazil, although there is a share of the total absenteeism related to shorter leaves that do not require public pensions.

- Lines 385-391: The Global Strategy considers a set of policy and interventions that should be implemented together (awareness and commitment, health services’ response, community action, drink-driving policies and countermeasures, availability of alcohol, marketing of alcoholic beverages, pricing policies, reducing the negative consequences of drinking and alcohol intoxication, reducing the public health impact of illicit alcohol and informally produced alcohol, and monitoring and surveillance. Besides the importance of taxation, for example, in Brazil part of the production of beers is subsidized in tax-free zones.

- This study may be an important tool for strengthening and prioritizing policies in this field. What other analyses and studies are important to better understand and communicate the burden of alcohol consumption?

Reviewer #2: Congratulations on the interesting paper. It adds value to both the policy makers in Brazil and other courniers with similar sociocultural cotext.

Few suggestions to improve the manuscript.

1. Please indicate the percentage share by the private sector for providing in patient and out patient in Brazil.

2. The study does not take into account the indirect costs incurred by premature mortality associated with alcohol abuse. In most of the studies done elsewhere this is a major component of the total cost. If the authors are ready to analyze this component too, it would add much value to the manuscript. Otherwise, please mention this as a limitation.

6. PLOS authors have the option to publish the peer review history of their article (what does this mean?). If published, this will include your full peer review and any attached files.

Reviewer #1: No

Reviewer #2: **Yes: **Nadeeka K Chandraratne

---

## [Author Response · Author response to Decision Letter 0]

28 Apr 2022

Dear Editor,

We thank you for the opportunity to have our manuscript PONE-D-21-23873, entitled "Direct and indirect costs attributed to alcohol consumption in Brazil, 2010 to 2018", evaluated for publication by Plos One. After analysing all the comments made, we are happy to implement the suggestions to improve the manuscript, as described below. 

Yours sincerely, 

The Authors

Table 1: Summary 

Questions Reviewer 1 Reviewer 2

Is the manuscript technically sound, and do the data support the conclusions? Yes Yes 

Has the statistical analysis been performed appropriately and rigorously? Yes Yes 

Have the authors made all data underlying the findings in their manuscript fully available? Yes Yes 

Is the manuscript presented in an intelligible fashion and written in standard English? Yes Yes 

Review Comments to the Author See below See below

Editor’s comments

Editor, comment 1: “1. Please ensure that your manuscript meets PLOS ONE's style requirements, including those for file naming. The PLOS ONE style templates can be found at 

Our response: The manuscript format has been checked and revised; and is in accordance with the PLOS ONE’s style. 

Editor, comment 2: “2. Thank you for stating the following financial disclosure: "Unfunded studies". At this time, please address the following queries: a) Please clarify the sources of funding (financial or material support) for your study. List the grants or organizations that supported your study, including funding received from your institution. b) State what role the funders took in the study. If the funders had no role in your study, please state: “The funders had no role in study design, data collection and analysis, decision to publish, or preparation of the manuscript.” c) If any authors received a salary from any of your funders, please state which authors and which funders. d) If you did not receive any funding for this study, please state: “The authors received no specific funding for this work.” Please include your amended statements within your cover letter; we will change the online submission form on your behalf.”

Our response: The authors received no specific funding for this work.

Editor, comment 3: “3. Please provide additional details regarding participant consent. In the Methods section, please ensure that you have specified (1) whether consent was informed and (2) what type you obtained (for instance, written or verbal). If your study included minors, state whether you obtained consent from parents or guardians. If the need for consent was waived by the ethics committee, please include this information.”

Our response: Our study used only aggregate information from government databases, which is in the public domain and offers no possibility of identifying individuals. On this basis, there is no need to obtain consent forms. 

Editor, comment 4: “4. Please include captions for your Supporting Information files at the end of your manuscript, and update any in-text citations to match accordingly. Please see our Supporting Information guidelines for more information: http://journals.plos.org/plosone/s/supporting-information.”

Our response: We also included captions for your supporting information files as suggested by the editor. Our supporting information files are: S1_Table_Costs_2010.pdf, S2_Table_Costs_2011.pdf until S9_Table_Costs_2018.pdf; and S10_Table_PAR-Prevalence-RR.

Editor, comment 5: “5. Please review your reference list to ensure that it is complete and correct. If you have cited papers that have been retracted, please include the rationale for doing so in the manuscript text, or remove these references and replace them with relevant current references. Any changes to the reference list should be mentioned in the rebuttal letter that accompanies your revised manuscript. If you need to cite a retracted article, indicate the article’s retracted status in the References list and also include a citation and full reference for the retraction notice.”

Our response: There is no retracted paper in our references.

Reviewer 1’s comments

Reviewer 1, comment 1: “The manuscript aims to estimate the direct costs and indirect costs attributed to alcohol consumption in Brazil between 2010 and 2018, considering the diseases for which alcohol is a risk factor.”

Our response: We would like to thank you for your consideration and time spent reviewing our manuscript.

Reviewer 1, comment 2: “Introduction. The introduction is complete in describing the health burden of alcohol consumption in the world, the trends in consumption in Brazil and other aspects of the issue. Nevertheless, it would benefit from addressing how alcohol, as a risk factor, is associated to the main health outcomes of excessive consumption.”

Our response: Thank you for your comment. We added this information as suggested by the reviewer. 

“There are several mechanisms by which alcohol consumption causes harms to drinkers. First, it is associated with deterioration of organs and tissues, causing diseases such as cancer, liver and heart diseases. Second, it causes alcohol dependence, which is associated with loss of self-control over volume and frequency of alcohol use. This condition also involves mental disorders, such as depression and psychoses. Third, it leads to intoxication, increasing the propensity of getting into dangerous situations under psychoactive effect5.

Reviewer 1, comment 3: “Additionally, what are the trends of alcohol consumption over time according to the national telephone survey, which is implemented yearly?”

Our response: Thank you for your comment. We have not identified any study that investigated the alcohol consumption trend over the time using VIGITEL database. Some studies have investigated the binge drinking over time, which has increased in the period from 2006 and 2018. 

Reference: Sanchez, Z.M. et al. Tendência do beber episódico excessivo nas capitais brasileiras e no Distrito Federal, 2006-2018: um estudo ecológico de séries temporais. Epidemiol. Serv. Saude, Brasília, 29(4):e2020078, 2020. 

Reviewer 1, comment 4: “Methods. The study uses a traditional cost-of-illness approach, using the population attributable risk associated with the prevalence of alcohol consumption and relative risks for different disease outcomes. It is not clear if the PAR are calculated for the entire population or if it is separated by age and sex groups. The disaggregation would be important to incorporate the differences in alcohol consumption among different population groups, such as those reported in the introduction (“consumption is highest among men, youth, people with higher education, and individuals who are single or divorced”).”

Our response: Thank you for raising this issue. We totally agree with the reviewer that stratification by sex and age groups certainly would add value to the analyses, since alcohol consumption varies according to these individual characteristics. However, to calculate the PAR we need the prevalence of alcohol consumption and the relative risk for each disease by which alcohol consumption if risk factor stratified by sex and age groups. Although there were data to calculate the prevalence of alcohol consumption by sex and age groups, the relative risk did not. On this basis, it was not possible to include these stratified analyses in the study. We acknowledged this information as limitation of the study. 

“Finally, our main source for relative risk from each disease by which alcohol consumption if risk factor did not provide data stratified by sex and age groups. On this basis, we were not able to provide cost estimated by sex and age groups.”

Reviewer 1, comment 5: “The PAR are based on a deterministic approach for the outputs. Nevertheless, estimations of the health and economic burden of disease or risk factors could use a probabilistic approach to incorporate the uncertainty of the inputs in the final results. Has this option been considered?”

Our response: Thank you for your comment. We have discussed the used of both deterministic and probabilistic approaches to handling uncertainties surrounding our data. Although we recognised the attractiveness of the latter, we opted for deterministic since it provides one-way sensitivity analyses by varying parameters according to the confidence intervals of each input (relative risk and prevalence). We judge the results from the one-way analyses would be relevant to the decision-makers.

Reviewer 1, comment 6: “Regarding the prevalence of alcohol consumption, why was the data from the 2013 National Health Survey included in the analysis? It may provide possible time changes in consumption in the period analyzed.”

Our response: Thank you for your comment. We used 2013 prevalence because it was the most reliable data available at the time we were conducting our study. As we provided sensitivity analyses based on the confident intervals of the alcohol consumption’s prevalence, it probably captured the changes in consumption mentioned by the reviewer. 

Reviewer 1, comment 7: “Why were the early retirements caused by diseases not included in the indirect costs? This data is usually available from the same sources as the sick leaves.”

Our response: Thank you for raising this issue. We have not requested data on early retirements, so we cannot affirm that they are available at the INSS database. Although we can do another request to INSS to ask data on early retirement (if available), it would take a couple of months to obtain them and another couple of months to analyse these data. On this basis, we acknowledged this information as limitation of the study (we also included premature deaths as suggested by reviewer 2).

 “Moreover, we have not included early retirements and premature deaths attributable to alcohol consumption, which is also an important source of indirect costs.”

Reviewer 1, comment 8: “Results. It is important to include the confidence intervals for the estimates, both in the tables and in the text.”

Our response: Thank you for the comment. The reporting of results from a cost-of-illness study is always a challenge due to the extensive number tables and figures. Based on that, we opted for including the information requested by the reviewer in the supplementary material. We added the confidence interval for the attributed costs by disease, type of cost and year. 

Reviewer 1, comment 9: “Would it be possible to express the standard deviation or standard error (or the IC 95%) for the prevalences, together with the IC for the relative risks?

Our response: Thank you for your comment. As suggested by the reviewer, we included IC 95% for relative risk and Population Attributable Risk in the supplementary material.

Reviewer 1, comment 10: “Discussion. Lines 290-304: are the results of this study on the hospitalization costs for cancer comparable to the study by Coutinho et al?”

Our response: Thank you for your comment. Coutinho et al (2016) used the same method (cost of illness with PAR) and database (inpatient and outpatient) to extract cost attributable to alcohol consumption in Brazil. However, we used different sources for prevalence and relative risk for diseases by which alcohol is risk factor. In our study, both prevalence and relative risks were stratified by levels of alcohol consumption (from zero to 72g/day and over, with frequency of 12g/day). Moreover, we included other types of cancers than Coutinho and colleagues, such as colon and rectum, lip and oral cavity cancers. Based on these differences, our results are not comparable. 

Reviewer 1, comment 11: “Line 310: The estimation of absenteeism is also underestimated because the sick pay leaves only include periods of 15 days or more away from the job paid by the INSS in Brazil, although there is a share of the total absenteeism related to shorter leaves that do not require public pensions.”

Our response: Thank for your comment. In fact, the reviewer is correct. We highlighted this information as a limitation of our study. 

“Another limitation is that the INSS data only records sick leave for formal workers in Brazil. In addition, only leave longer than 15 days is registered with the INSS, in case of shorter leave, the employer is responsible for the costs, and there are no records of these expenses. Moreover, we have not included early retirements attributable to alcohol consumption, which is also an important source of indirect costs. All these limitations tend to reduce the picture of the real economic burden of alcohol in Brazil.”

Reviewer 1, comment 12: “Lines 385-391: The Global Strategy considers a set of policy and interventions that should be implemented together (awareness and commitment, health services’ response, community action, drink-driving policies and countermeasures, availability of alcohol, marketing of alcoholic beverages, pricing policies, reducing the negative consequences of drinking and alcohol intoxication, reducing the public health impact of illicit alcohol and informally produced alcohol, and monitoring and surveillance. Besides the importance of taxation, for example, in Brazil part of the production of beers is subsidized in tax-free zones.”

Our response: Thank you for your comment. We totally agree with the reviewer that we need multidimensional interventions to tackle alcohol consumption and taxation is just one among many others needed. 

Reviewer 1, comment 13: “This study may be an important tool for strengthening and prioritizing policies in this field. What other analyses and studies are important to better understand and communicate the burden of alcohol consumption?”

Our response: Thank for your comment. We suggested new studies on this topic as required by the reviewer. 

“There are few cost studies related to alcohol produced in Brazil. Thus, the production of knowledge in this area is necessary. Understanding the economic impacts due to alcohol consumption is essential to scale this public health problem in all its facets, in addition to the morbidity and mortality it causes. There is also need for evidence on disease burden of alcohol consumption on other diseases than ones included in our study, including mortality outcome. It is also important to investigate the barriers and facilitators of implementing policies against alcohol consumption among age groups, since they may respond differently depend on the strategy adopted.”

Reviewer 2’s comments

Reviewer 2, comment 1: “Congratulations on the interesting paper. It adds value to both the policy makers in Brazil and other courniers with similar sociocultural cotext.”

Our response: We would like to thank you for your consideration and time spent reviewing our manuscript.

Reviewer 2, comment 2: “Few suggestions to improve the manuscript. 1. Please indicate the percentage share by the private sector for providing in patient and out patient in Brazil.”

Our response: Thank you for this comment. We added this percentage as suggested by the reviewer. 

“About 25% of the Brazilian population have access to private health insurances, which was not cover by our study.”

Reviewer 2, comment 3: “2. The study does not take into account the indirect costs incurred by premature mortality associated with alcohol abuse. In most of the studies done elsewhere this is a major component of the total cost. If the authors are ready to analyze this component too, it would add much value to the manuscript. Otherwise, please mention this as a limitation.”

Our response: Thank you for raising this important issue. In fact, premature mortality is one of the main costs related to alcohol consumption, particularly for young men who were involved with traffic accidents caused by drinking and driving episodes. We did not include these costs because we have opted for using only real-world costs incurred by the Ministry of Health (healthcare costs) and Ministry of Labour and Social Security (sick leaves costs), i.e., money that in fact both ministries have spent owing to alcohol consumption. We acknowledged this information in the limitation of the study (we also included early retirement as suggested by reviewer 1). 

“Moreover, we have not included early retirements and premature deaths attributable to alcohol consumption, which is also an important source of indirect costs.”

---

## [Decision Letter · Decision Letter 1]

6 Jun 2022

Direct and indirect costs attributed to alcohol consumption in Brazil, 2010 to 2018

PONE-D-21-23873R1

Dear Dr. de Freitas,

We’re pleased to inform you that your manuscript has been judged scientifically suitable for publication and will be formally accepted for publication once it meets all outstanding technical requirements.

Kind regards,

Roy Cerqueti, Ph.D.

Academic Editor

PLOS ONE

Additional Editor Comments (optional):

Reviewers' comments:

Reviewer's Responses to Questions

**Comments to the Author**

1. If the authors have adequately addressed your comments raised in a previous round of review and you feel that this manuscript is now acceptable for publication, you may indicate that here to bypass the “Comments to the Author” section, enter your conflict of interest statement in the “Confidential to Editor” section, and submit your "Accept" recommendation.

Reviewer #1: All comments have been addressed

Reviewer #2: All comments have been addressed

2. Is the manuscript technically sound, and do the data support the conclusions?

Reviewer #1: Yes

Reviewer #2: (No Response)

3. Has the statistical analysis been performed appropriately and rigorously? 

Reviewer #1: Yes

Reviewer #2: (No Response)

4. Have the authors made all data underlying the findings in their manuscript fully available?

Reviewer #1: Yes

Reviewer #2: (No Response)

5. Is the manuscript presented in an intelligible fashion and written in standard English?

Reviewer #1: Yes

Reviewer #2: (No Response)

6. Review Comments to the Author

Reviewer #1: All comments from the reviewers have been addressed.

The study contributes to the knowledge in the field of cost of disease and risk factors in Brazil and the methodologies can be useful in other settings.

Reviewer #2: (No Response)

7. PLOS authors have the option to publish the peer review history of their article (what does this mean?). If published, this will include your full peer review and any attached files.

Reviewer #1: No

Reviewer #2: **Yes: **Nadeeka Chandraratne

---

## [Editor Report · Acceptance letter]

5 Oct 2022

PONE-D-21-23873R1 

Direct and indirect costs attributed to alcohol consumption in Brazil, 2010 to 2018 

Dear Dr. de Freitas:

I'm pleased to inform you that your manuscript has been deemed suitable for publication in PLOS ONE. Congratulations! Your manuscript is now with our production department. 

Kind regards, 

on behalf of

Professor Roy Cerqueti 

Academic Editor

PLOS ONE